# Epidemiologic, clinical, and therapeutic aspects of formally identified *Echis romani* bites in northern Cameroon

Jean-Philippe Chippaux[1]*, Pierre Amta[2], Yoann Madec[3], Rodrigue Ntone[4], Gaëlle Noël[5], Pedro Clauteaux[5], Yap Boum II[6], Armand S. Nkwescheu[7], Fabien Taieb[8]

**1** Université Paris Cité, IRD, Inserm, MERIT, Paris, France, **2** Tokombéré Hospital, Tokombéré, Mora, Cameroon, **3** Emerging Diseases Epidemiology Unit, Institut Pasteur, Paris Cité University, Paris, France, **4** Epicentre Yaoundé, Yaounde, Cameroon, **5** Institut Pasteur, Translational Research Center, Paris Cité University, Paris, France, **6** Institut Pasteur de Bangui, Bangui, Central African Republic, **7** Cameroon Society of Epidemiology, Yaounde, Cameroon, **8** Institut Pasteur Medical Center, Paris Cité University, Paris, France

* jean-philippe.chippaux@ird.fr

## Abstract

### Background

Species of the genus *Echis*, particularly those of the '*ocellatus*' group, are responsible for the majority of snakebite envenomations and deaths in the savannas of sub-Saharan Africa. In a clinical study conducted in Cameroon, we treated a series of patients bitten by formally identified *E. romani*. The clinical outcomes are described and discussed.

### Methodology/Principal findings

Specimens brought in by the victim were identified by a herpetologist. Clinical description and therapeutic management followed a standardized protocol applied by trained physicians.

We included 92 patients, 95% of whom (n = 87) were envenomated. More than one third of the bites occurred during agricultural work, and one quarter in the victim's home. The bite site was the foot in 48 victims (52%) and the hand in 40 others (43%), mostly children and teenagers. Cytotoxic syndrome was observed in 84 of the 87 envenomated patients (97%). Hemostasis disorders were observed in 78 patients (90%), 38 of whom (44%) experienced bleeding during hospitalization. In 5 of the latter (13%), the bleeding recurred, whereas it had stopped after antivenom administration. A further 7 patients, who were not bleeding on arrival, experienced late bleeding despite antivenom administration. Four patients (4.3%), including one pregnant woman, died. All were bleeding on arrival. Finally, 2 patients (2.2%) had permanent sequelae of moderate severity.

**Data availability statement:** The clinical study data have been deposited in a publicly open accessible repository and are available at https://doi.org/10.5281/zenodo.15435507

**Funding:** The study was funded by the Institut Pasteur. Inosan Biopharma contributed to the financing through a grant paid to the Institut Pasteur. Inosan Biopharma had no role in study design, data collection and analysis, decision to publish, or preparation of the manuscript. None of the co-authors received grant or personal funding from any source.

**Competing interests:** The authors have declared that no competing interests exist.

## Conclusion/Significance

This study confirms the frequency and severity of hemorrhagic complications in *E. romani* envenomation. Lethality remains high despite antivenom treatment. Cytotoxic syndromes, present in 95% of victims, rarely progress to extensive necrosis.

## Author summary

The *Echis* genus is responsible for the majority of envenomations and deaths in the savanna and grassland regions of sub-Saharan Africa. The authors examined and treated 92 patients bitten in northern Cameroon by *E. romani* –a species of the 'ocellatus' group– formally identified from the specimen brought by the victim. Five of them showed no symptoms of envenomation (5% dry bites). Of the 87 envenomated patients, 84 (97%) had a cytotoxic syndrome characterized by edema of varying severity. Hemostasis disorders were observed in 78 patients, 38 (49%) of whom experienced bleeding during hospitalization. Among the latter, 5 patients (13%) showed recurrence of bleeding after cessation of bleeding. Seven patients who were not bleeding on arrival experienced late bleeding despite antivenom administration. Four patients, all with bleeding, including one pregnant woman, died. Two others had permanent sequelae of moderate severity. This study confirms that *E. romani* is a major threat to the rural population of northern Cameroon due to hemorrhagic complications and, secondarily, the risk of motor disability from tissue necrosis.

## Introduction

In sub-Saharan Africa (SSA), more than 300,000 snakebite envenomations (SBEs) are treated in health facilities each year. Approximately 10,000 deaths and as many permanent disabilities are reported [1]. Most SBEs occur in rural areas during agricultural work among poor populations [2]. However, these figures probably represent less than one third of the reality [1,3].

In the savannas of SSA, north of the equator, species of the genus *Echis* account for more than two thirds of SBE [4–11]. Most of these species belong to the *Echis ocellatus s.l.* group, which consists of 3 species: *E. jogeri* in eastern West Africa (Senegal, Mali, Guinea), *E. ocellatus s.s.* (from Mali and Guinea to western Nigeria), and *E. romani* (from central Nigeria to southern Sudan) (Fig 1) [12,13]. Until 2018, they were considered to belong to the same species, *E. ocellatus*, which was previously a subspecies of *E. carinatus* before being elevated to species status in 1976 [14–16].

Analysis of *E. ocellatus* venom has identified about 35 proteins belonging to 8 different families [17,18]. Some are particularly abundant: metalloproteinases (SVMPs) make up 67% of the venom, phospholipases $A_2$ ($PLA_2$s) for 13%, disintegrins and C-type lectins for 7% each. There are also half a dozen proteases, each in concentrations of no more than 5%.

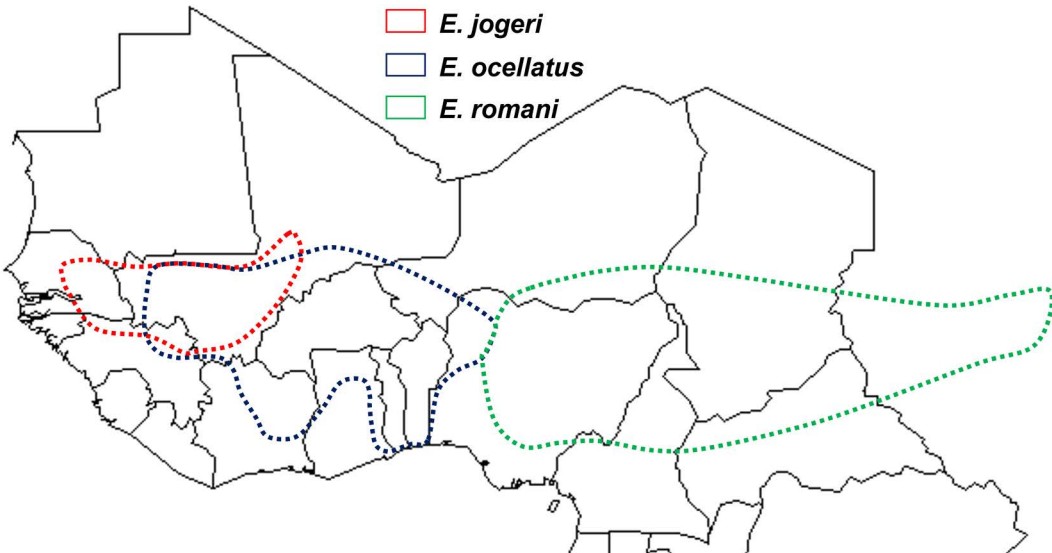

**Fig 1. Geographic distribution of the 3 species of the *Echis ocellatus* group in West and Central Africa (original map hand-drawn [15] based on data from [12,13,15]; this original map is compatible with a CC BY 4.0 License).**

Snake venom SVMPs are classified into four main groups according to their structure, with different pharmacological effects. Group P-III SVMPs, which account for nearly 40% of *E. ocellatus* venom, are the most hemorrhagic [17–20]. They share a 91% sequence similarity with ecarin, a prothrombin activator from *E. carinatus* and *E. pyramidum*, that induces severe consumptive coagulopathy [21]. Group P-IV SVMPs (more than 20% of *E. ocellatus* venom) bear a strong resemblance to echicetin, a C-type lectin present in *Echis* venom, that binds to the GPIb glycoprotein complex and inhibits platelet aggregation [22]. Depending on their group, SVMPs a) activate prothrombin to generate meizothrombin, which is responsible for an unstable, transient, weakly thrombogenic clot [23,24], b) directly digest fibrinogen/fibrin, c) inhibit platelet aggregation and d) degrade the extracellular matrix and basement membrane of blood vessels [19,24,25–27]. In fact, the studies by Wagstaff et al. and Hasson et al. were performed on venom from *Echis* sp. from Kaltango, northeastern Nigeria [17,21]. Therefore, they are probably belonging to the *E. romani* species.

PLA$_2$s are involved in several coagulation factors, in particular by hydrolyzing membrane phospholipids and those of the prothrombinase complex (factor X) and by inhibiting platelet aggregation [28–30].

Involved in the regulation of cell adhesion, disintegrins, specific integrin antagonists, and C-type lectins inhibit platelet aggregation, and prevent blood clot formation.

In Cameroon, *E. romani* is abundant north of the Adamaoua Plateau [31,32]. As part of the clinical study "Evaluation d'un Sérum Antivenimeux en Afrique" ("Antivenom evaluation in Africa") (ESAA) conducted in Cameroon between 2019 and 2021 [33,34], we were able to identify 151 snakes responsible for bites and SBEs, including 95 *E. romani* [35].

However, snakes responsible for bites are rarely identified, because few are brought to the hospital and health workers are not trained to identify them. The aim of the present work was to describe the epidemiology, clinical symptoms and severity of *E. romani* bites, and their evolution after treatment with a polyvalent antivenom marketed in Cameroon.

## Methods

### Ethics statement

The study was conducted in accordance with the Declaration of Helsinki, and approved by the Institut Pasteur Institutional Review Board (Paris, N° IRB00006966/2017-06) and the National Ethics Committee of Cameroon (Yaoundé,

---

N°2018/03/994/CE/CNERSH/SP). The study was authorized by the Ministry of Public Health of Cameroon (N°631-18.18). Informed consent was obtained from all subjects involved in the study.

This is an ancillary observational study based on the ESAA study, the main data of which have been published elsewhere [33,34]. It focuses on patients bitten by *E. romani*. The ESAA study was a prospective, real-life clinical investigation. The study population and the methods used for data collection and analysis have been described in detail elsewhere [33–35]. They are briefly reviewed here.

Patients were recruited between October 25, 2019 and May 3, 2021, at fourteen health centers across Cameroon. We restricted the data used in this study to those collected at the seven health centers located in northern Cameroon where *E. romani* is present (Fig 2).

Inclusion criteria were: (a) snakebite, with or without SBE, (b) age greater than or equal to five years, (c) no known allergy to antivenom of equine origin, (d) no administration of antivenom prior to hospital admission, and (e) signature of informed consent.

After enrollment, assessment of SBE severity was based on measurement of clinical grades for cytotoxicity, bleeding and neurotoxicity. Grades for cytotoxicity (assessed by pain intensity and edema size), bleeding (assessed by the location and extent of bleedings) and neurotoxicity (based on the degree of cranial nerve paralysis) are described in S1 Appendix. A coagulation test (Whole Blood Clotting Test 20 minutes or 20WBCT) was performed for each patient to assess in

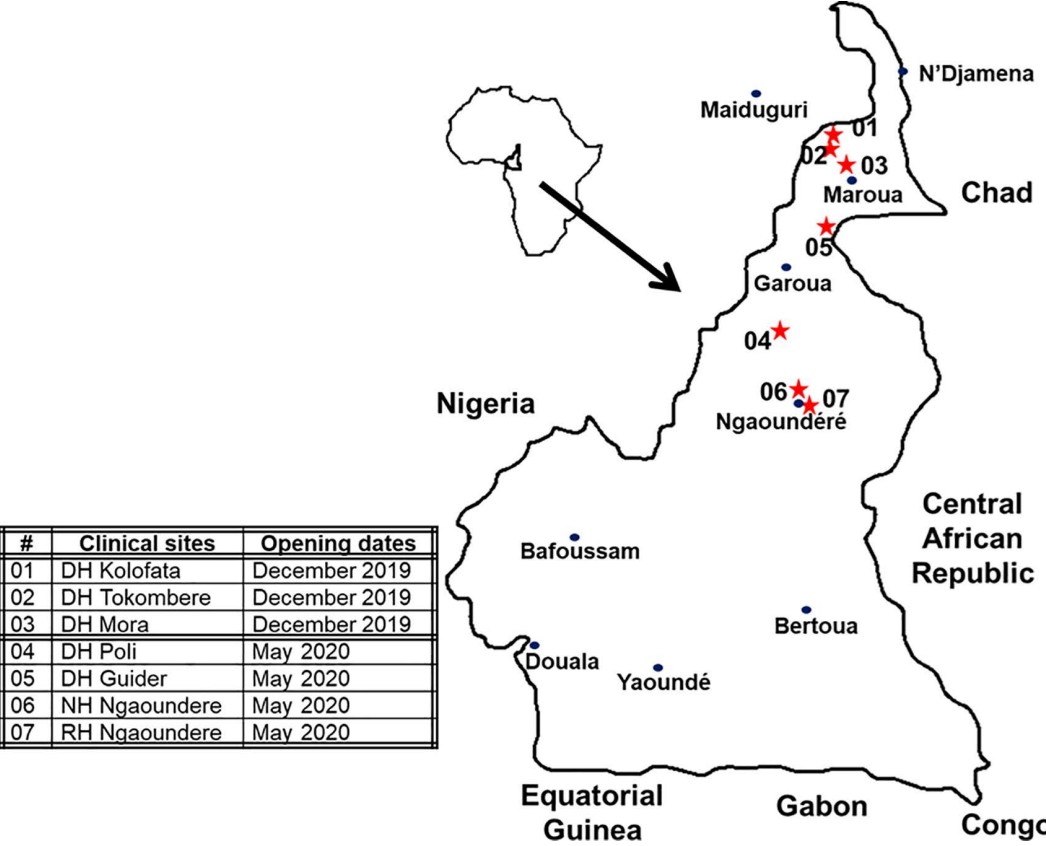

| # | Clinical sites | Opening dates |
|----|---------------|---------------|
| 01 | DH Kolofata | December 2019 |
| 02 | DH Tokombere | December 2019 |
| 03 | DH Mora | December 2019 |
| 04 | DH Poli | May 2020 |
| 05 | DH Guider | May 2020 |
| 06 | NH Ngaoundere | May 2020 |
| 07 | RH Ngaoundere | May 2020 |

**Fig 2. Location of centers participating in the ESAA study within the range of E. romani (01-03: Far North (Sahel); 04-07: North (Savannah)).**
(Map redrawn by hand from Wikipedia. https://fr.wikipedia.org/wiki/Cameroun#/media/Fichier:Cameroon_sat.png, CC0 1.0, accessed on December 5, 2024).

vitro blood coagulability. It was measured by checking whether blood drawn from a dry tube clotted within 20 minutes (S1 Appendix).

The decision to administer the antivenom was made by the physician-investigator based on the therapeutic algorithm recommended by the Cameroon Ministry of Health and the manufacturer's guidelines (S2 Appendix). A dry bite is a bite from a snake that has been identified as venomous without causing clinical symptoms. These victims did not receive antivenom.

We used Inoserp PAN-AFRICA (IPA), the reference antivenom in Cameroon. This is a lyophilized polyvalent antivenom composed of highly purified immunoglobulin fragments prepared by immunizing horses with the venoms of fourteen snake species (*Echis ocellatus s.l.*, *E. pyramidum*, *E. leucogaster*, *Bitis gabonica*, *B. nasicornis*, *B. arietans*, *Naja haje*, *N. melanoleuca*, *N. nigricollis*, *N. pallida*, *Dendroaspis polylepis*, *D. viridis*, *D. angusticeps*, et *D. jamesoni*). Each vial contains no more than 1 g of total proteins and neutralize at least 250 $LD_{50}$ of the venom of *E. ocellatus*, *B. arietans*, *N. nigricollis* and *D. polylepis*. IPA from a single batch (#8IT11001; expiration date Nov. 2021) was used for all patients at all study centers.

IPA, which was provided free of charge to all study patients, was administered intravenously: 2 vials for cytotoxicity and/or hemorrhage or 4 vials for neurotoxicity, and repeated every 2 hours in patients with onset, persistence, or worsening of bleeding or neurotoxic disorders (S2 Appendix).

Data were collected on case report forms and entered into REDCap data collection software versions 13.7.1 (Vanderbilt University, Nashville, TN, USA) [36,37]. Sociodemographic data, circumstances of snakebite, time and clinical presentation at hospital admission were collected. Systematic evaluations were also performed at 12, 24, 48 hours and 3 days after the first injection. At these timepoints, IPA injection was repeated at the same dose in patients presenting persistence or worsening of bleeding disorders, and patients were clinically evaluated 2 hours after each IPA administration [see for details 33]. An evaluation was also conducted at hospital discharge, and again approximately 15 days after hospitalization, to assess late tolerance (serum sickness) and to look for possible sequelae [see for details 34].

The snake responsible for the bite was formally identified when it was brought to hospital by the patient or his family, or when a photograph was taken. Photos of the snake (Fig 3) were sent to an expert (J.-P. Chippaux) to determine the species using keys and published descriptions [15,16].

Efficiency was evaluated in all patients with SBE symptoms who received at least one injection of IPA and underwent at least one post-injection clinical evaluation.

Efficacy was assessed by evaluating the speed of resolution of SBE symptoms during hospitalization. This analysis was performed in patients with edema and/or bleeding. Patients with only an abnormal 20WBCT without bleeding were not included, because 20WBCT was not routinely performed beyond 2 hours after the initial injection.

Analyses were performed using Stata 17 (Stata Corp., College Station, TX, USA).

The ESAA study was registered on clinicaltrial.gov (NCT03326492). Ethical clearance was obtained from the Institut Pasteur Institutional Review Board (Paris, N° IRB00006966/2017-06) and the National Ethics Committee of Cameroon (Yaoundé, N°2018/03/994/CE/CNERSH/SP). The study was approved by the Ministry of Public Health of Cameroon (N°631-18.18).

Formal written informed consent was obtained from all patients enrolled in the study, including children for whom written consent was obtained from the parent/guardian accompanying the child to the hospital. Patients admitted to the study centers who did not consent to participate in the study were treated according to the national protocol. No data were collected on these patients.

A scientific committee met twice to review the results, with particular focus on serious adverse events. This committee included the principal investigators from Paris and Yaoundé, a representative of the sponsor (Institut Pasteur, Paris) and two independent international experts. In some cases, other independent experts were consulted from time to time.

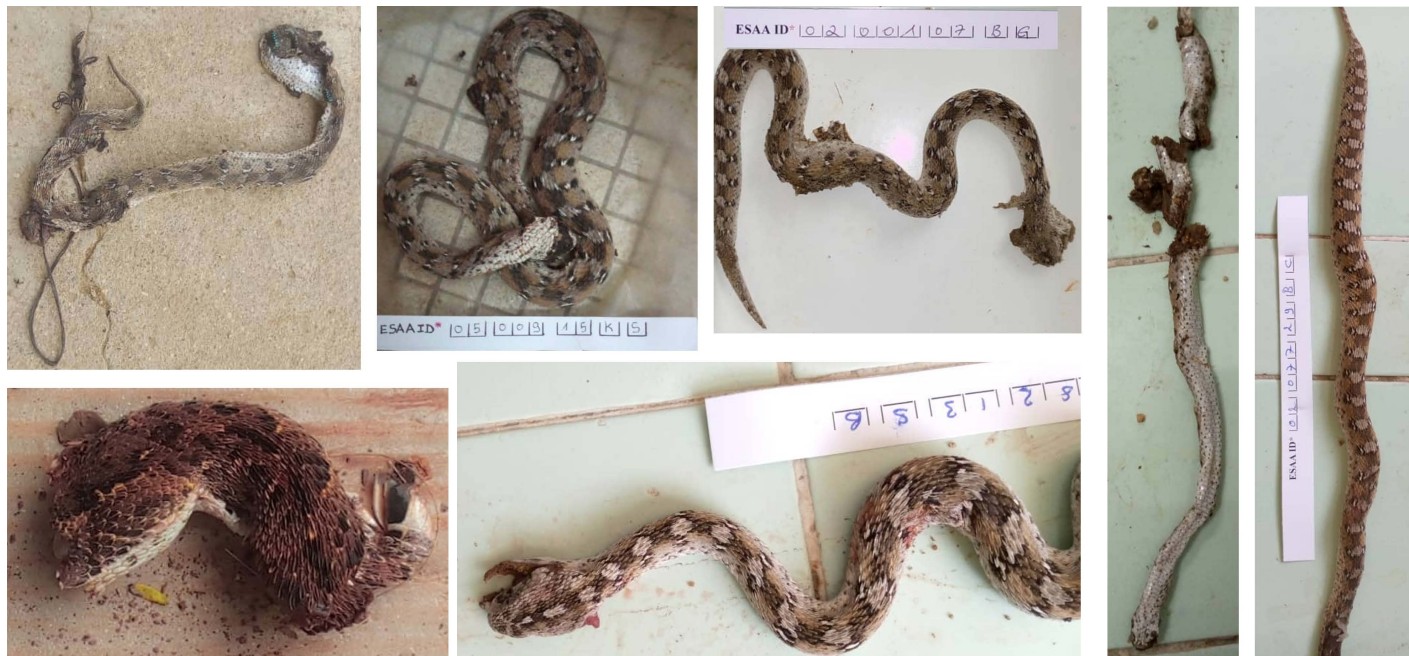

**Fig 3. Several examples of photographs of *Echis romani* sent in for identification.** Some showed severely damaged specimens. However, the number of rows of keeled dorsal scales, the dorsum with bright ivory spots outlined in dark brown, and the pale cream venter mottled with black spots are sufficient for species identification compared to confusing other species (Photo: ESAA Program, Institut Pasteur).

## Results

Of 159 snakes brought in by snakebite victims, 95 were formally identified as *E. romani*. Of these, 92 were included in the study. Five patients (5.4%) were asymptomatic and did not receive IPA, but may have received symptomatic treatment to reduce their anxiety (Fig 4).

The sex ratio was 1.04 (47 males versus 45 females). The median age was 25 years [IQR: 14–36; range: 5–70]. The frequency of bites at each study site is shown in Fig 5. The snakebite occurred between 6am and 7pm, in daylight, in 69 patients (75%). A higher incidence of *E. romani* bites was observed at the end of the long dry season (March-April) and during the two rainy seasons between July and October (Table 1).

More than 2/3 of the snakebites occurred in the fields or while travelling, especially to get to the fields (Table 2). Of the former, 34/92 bites were directly related to farm work, i.e., 37% of all bites. Almost a quarter of the bites occurred at home, either while sleeping, or during household or leisure activities.

The majority of bites occurred on the foot (n = 48; 52%). Other bites were on the hand (n = 40; 43%), forearm (n = 2; 2%), pelvis in one patient passing to the toilet and head in another who had been herding sheep. Hand bites occurred predominantly in children under 16 years of age (n = 19 out of 28; 68%) compared with 21 of the 64 adult patients (33%; p = 0.004).

The median time between the bite and hospital presentation was 3h30mns [IQR: 2h07mn–7h71mn; range: 0–148h].

Of the 92 victims included in the study, 87 (95%) showed SBE symptoms and received IPA injection.

A cytotoxic syndrome (sharp pain, extensive edema, sometimes a skin lesion prefiguring necrosis) was observed in 84 (96.6%) of the envenomed patients, 9 of whom had no other symptoms. In the 50 (57.5%) patients who reported pain at the bite site, the median patient-rated intensity was 6 [IQR: 4–7]. In the 84 patients with edema, the median grade was 2 [IQR: 1–2].

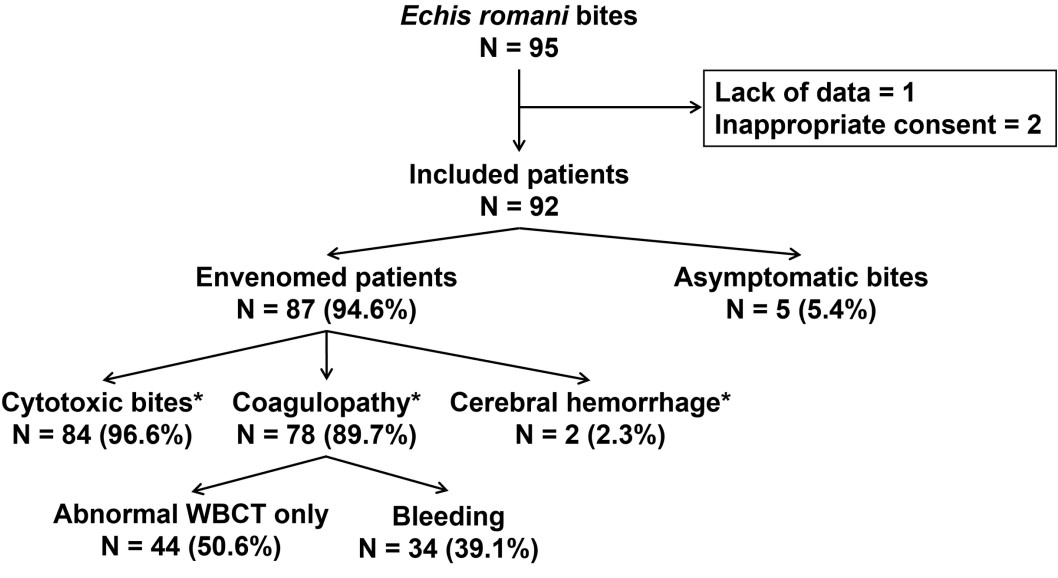

**Fig 4. Flowchart of patient inclusion and clinical presentation on admission.**

The 20WBCT was abnormal in 78 (90%) patients with SBE, and 31 (36%) patients had bleeding at inclusion, 2 of whom had neurological disorders related to intracerebral bleeding that could not be confirmed by medical imaging. In 7 patients (8%) who were not bleeding on arrival, hemorrhage occurred in the hours following admission despite the administration of IPA. In total, therefore, 38 patients (43.7% of SBE) experienced bleeding after being bitten by *E. romani*. In addition, five patients (5.7%) whose bleeding had stopped experienced a recurrence of bleeding. The median time to recurrence of bleeding was 5h00mn (IQR: 2h30mn–6h; range: 2–48). The median time from bite to hospital arrival was 3.3 hours [IQR: 2.1–6] for patients not bleeding on arrival and 5.3 hours [IQR: 2–12.4] for patients who bleed at any time during hospitalization (p = 0.16).

There was no difference in the incidence of bleeding or its characteristics according to gender, age, season of onset or circumstances of the bite (p > 0.05).

The 87 SBE patients received IPA in combination with symptomatic treatment, which varied from case to case and from health center to health center (analgesic, anti-inflammatory, blood transfusion, antibiotics).

In all patients, the median dose of antivenom was 2 vials [IQR: 2–4; range: 2–10] (Fig 6). In patients with bleeding, the median dose (4 vials [IQR: 2–4; range: 2–10]) was significantly different from that in patients without bleeding (p = $10^{-6}$).

The efficacy of IPA on edema was modest, with a median time to resolution of 72 hours [IQR: 65.25–72.2; range: 1–>120.2]. In contrast, the median time taken to stop bleeding was 4.33 hours [IQR: 1.83–56.6; range: 0.78–99.9] (Fig 7).

No serious adverse events (i.e., angioedema and bronchospasm) were reported in the 87 patients who received IPA and underwent clinical evaluation. No urticaria was observed, but 3 patients experienced pruritus (2 within 2 hours and 1 more than 2 hours after IPA administration), and another showed diffuse erythema within 2 hours after IPA administration.

In terms of causality assessment, no patient experienced adverse events that were considered probably related to IPA administration according to the Naranjo algorithm (see Reference 34 for more methodological details). The three patients with pruritus and the patient with diffuse erythema were considered possibly related to IPA administration.

Four deaths (4.3%) occurred (Table 3). None of them showed symptoms of antivenom intolerance. Patient 1 died at home and we had no relevant information to determine the cause of death. The other three died of hemorrhagic

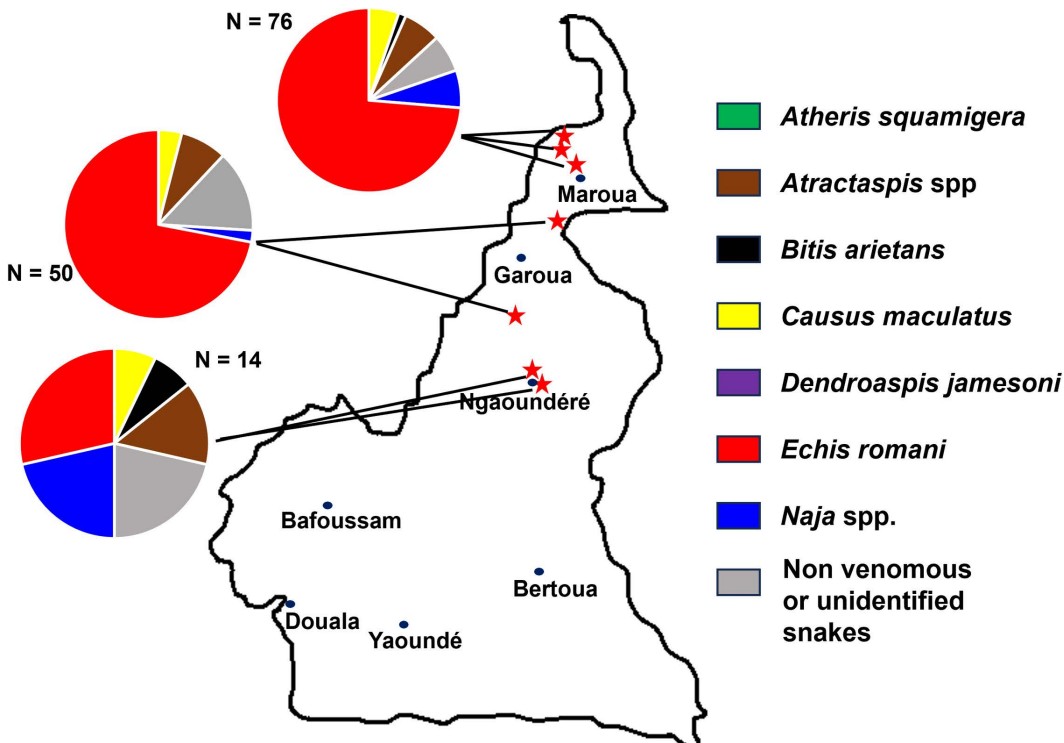

**Fig 5. Frequency of *E. romani* bites (n = 92) in the different study regions.** (Map redrawn from Wikipedia. https://fr.wikipedia.org/wiki/Cameroun#/media/Fichier:Cameroon_sat.png, CC0 1.0, accessed on 5 December 2024).

**Table 1. Seasonal incidence of *E. romani* bites (n = 92).**

| Month | # snakebites | Males | Females | Children | Adults | Envenomed | Dry snakebites |
|---|---|---|---|---|---|---|---|
| January | 6 | 4 | 2 | 4 | 2 | 6 | 0 |
| February | 2 | 1 | 1 | 1 | 1 | 2 | 0 |
| March | 11 | 4 | 7 | 1 | 10 | 11 | 0 |
| April | 14 | 6 | 8 | 6 | 8 | 14 | 0 |
| May | 2 | 2 | 0 | 0 | 2 | 2 | 0 |
| June | 2 | 1 | 1 | 1 | 1 | 2 | 0 |
| July | 13 | 7 | 6 | 4 | 9 | 13 | 0 |
| August | 13 | 10 | 3 | 2 | 11 | 11 | 2 |
| September | 10 | 4 | 6 | 3 | 7 | 9 | 1 |
| October | 5 | 4 | 1 | 2 | 3 | 5 | 0 |
| November | 7 | 2 | 5 | 1 | 6 | 6 | 1 |
| December | 7 | 2 | 5 | 3 | 4 | 6 | 1 |

complications. Patient 76 suffered a cerebral hemorrhage and died less than an hour after arrival at the hospital. A pregnant woman (patient no. 75) had a hemorrhage during delivery. The vaginal delivery was normal, with the birth of a healthy, live baby. She was given 10 vials of antivenom and 5 bags of whole blood. After cessation of the hemorrhagic syndrome stopped, uterine and vaginal bleeding resumed, associated with anuria which preceded death. Finally, patient

**Table 2. Location and activity of the victims (n = 92) at the time of the bite.**

| Activity | # snakebites | Males | Females | Children | Adults | Envenomed | Bleeding | Dry snakebites |
|---|---|---|---|---|---|---|---|---|
| Agricultural work | 46 | 22 | 24 | 13 | 33 | 43 | 19 | 3 |
| Walking | 14 | 8 | 6 | 3 | 11 | 13 | 7 | 1 |
| Inside compounds | 10 | 5 | 5 | 0 | 4 | 3 | 1 | 1 |
| Sleeping | 10 | 2 | 8 | 1 | 9 | 10 | 2 | 0 |
| Passing to the toilets | 2 | 0 | 2 | 0 | 2 | 2 | 1 | 0 |
| Hunting | 5 | 5 | 0 | 4 | 1 | 5 | 2 | 0 |
| Collecting wood | 3 | 3 | 0 | 2 | 1 | 3 | 2 | 0 |
| Herding | 2 | 2 | 0 | 1 | 1 | 2 | 2 | 0 |

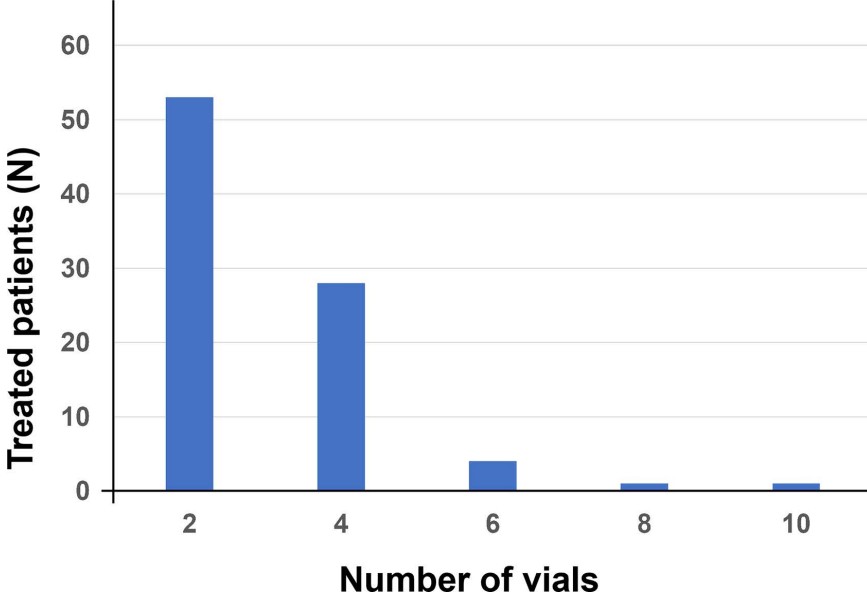

**Fig 6. Distribution of number of antivenom doses administered to patients (n = 87).**

no. 79 had a co-morbidity (malaria) and may not have received an adequate dose of antivenom, according to the scientific committee and the experts who reviewed this file.

In addition, two patients (2.3%) had permanent sequelae of mild to moderate severity despite rapid treatment (less than 5 hours) (Table 4). Patient 15 was not bleeding on arrival at the hospital but developed local bleeding (grade 1) 24 hours after admission. The bleeding stopped less than two hours after a second dose of antivenom was administered. The finger showed ankylosis of the joint of the last phalanx with deviation of the index finger (Fig 8). The second patient (# 80) presented with a grade 3 hemorrhagic syndrome on admission, which decreased to grade 1 two hours after antivenom administration and stopped before the fourth hour. The inflammatory syndrome and necrosis necessitated amputation of the right ring finger (Fig 9).

## Discussion

The main features of this study were formal identification of the snake responsible for the bite (*E. romani*), monitoring of symptoms at short intervals throughout the hospital stay, standardized management and late examination of the patient

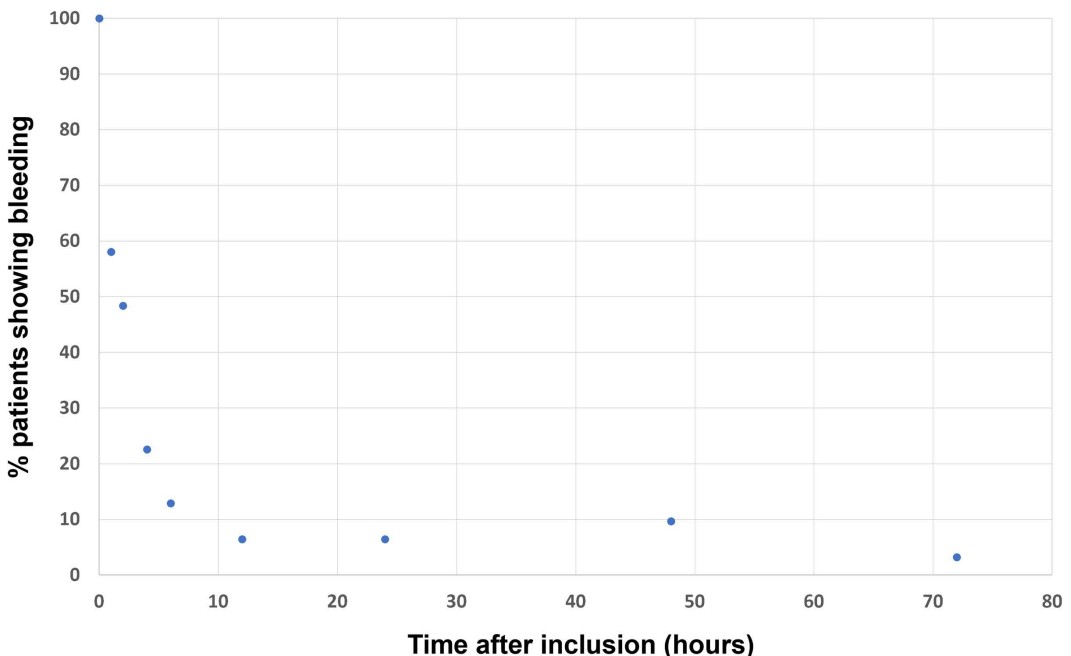

**Fig 7. Time to stop bleeding (N = 38) in patients receiving IPA.**

**Table 3. Death characteristics of 4 patients.**

| ID | Local-ity | Place of death | Gen-der[1] | Age | Time to treatment | Time to death | AV dose | Cause of death |
|---|---|---|---|---|---|---|---|---|
| 65 | Poli | Home* | M | 45 | 3 h | 168 h | 6 vials | Anemia + malaria |
| 75 | Poli | Hospital (Garoua)§ | F | 25 | 1.7 h | 114.3 h | 10 vials | Acute renal failure + obstetric hemorrhage due to envenomation |
| 76 | Poli | Hospital | M | 8 | 62.5 h | 63.5 h | 2 vials | Brain hemorrhage |
| 79 | Guider | Hospital | M | 12 | 1 or 2 h | 6/7 h | 4 vials | Hemorrhagic syndrome + malaria + insufficient antivenom dose |

[1]: M = male; F = Female;

*Discharged against medical advice; § = After emergency transfer due to deteriorating medical condition.

**Table 4. Definitive sequelae observed in 2 patients.**

| ID | Locality | Gender[1] | Age | Time to treatment | Number of vials | Type of sequelae |
|---|---|---|---|---|---|---|
| 15 | Tokombéré | M | 18 | 3h15 | 4 | Deformity and ankylosis of the index finger |
| 80 | Guider | M | 15 | 4h20 | 2 | Ring finger amputation |

[1]: M = male.

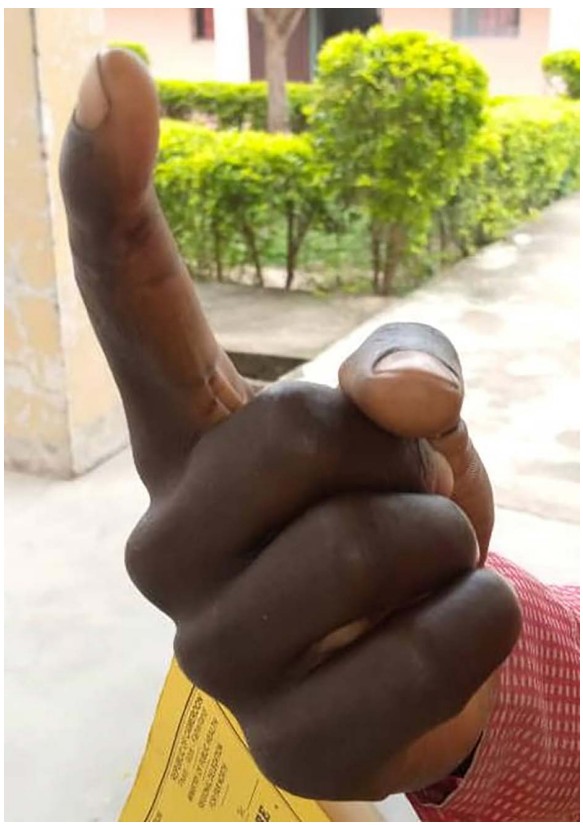

**Fig 8. Ankylosis of the last phalangeal joint with deviation of the right index finger (Photo: ESAA Program, Institut Pasteur).**

(beyond the second week) to assess late tolerance and sequelae of SBE [34]. A major limitation of the study was the lack of additional investigations after the initial examination, including 20WBCT, due to a lack of appropriate resources. Blood tests could have provided a better understanding of the causes of the hemorrhagic syndrome so common in SBE by vipers of the genus *Echis*. However, the clinical study that allowed us to carry out this ancillary study was designed as a real-life study, which limited the use of additional biological tests that are not readily available in peripheral health centers [33].

*Echis* of the 'ocellatus' group –and *E. romani* is no exception– are widespread in the Sudanese and Guinean savannas, particularly in rural areas (Fig 1). They are found in natural environments, but also in different types of plantations, especially village food plantations and cotton plantations [38]. They can even be considered part of the liminal fauna, present in concessions and sometimes in houses on the outskirts of towns and, more exceptionally, in the center of certain towns [4,39,40]. *Echis* is nocturnal. However, 69 of the 92 bites (75%) occurred during the day, suggesting that the snake was disturbed. The ecology of this species is unfortunately very poorly known and should be studied further in order to develop appropriate prevention measures.

The profile of the victims is similar to that described in most studies of *E. ocellatus* [4,31,38,41,42]. These are young patients bitten during agricultural and pastoral activities. Usually the sex ratio is unbalanced, unlike in our series. Males generally represent the majority of snakebites in SSA [1]. The high incidence of hand bites, particularly in children and teenagers under 16 years of age, can be explained by their habit of searching burrows with their bare hands in to

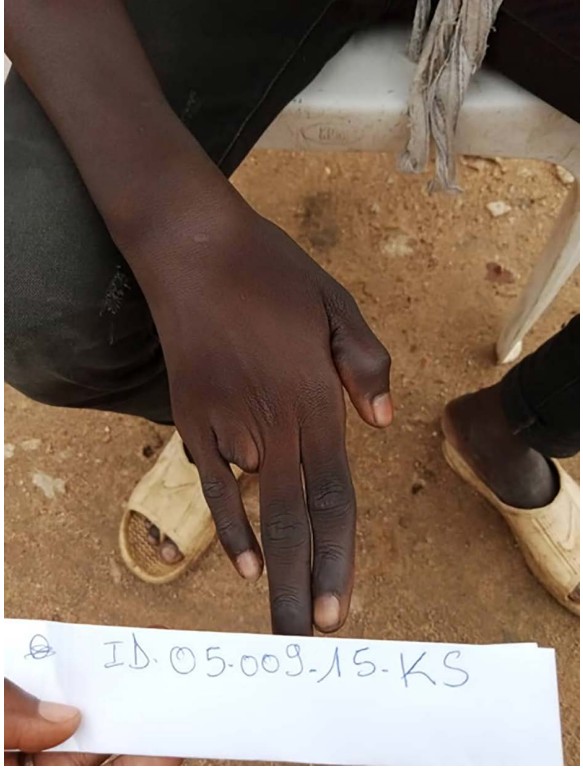

**Fig 9. Total amputation of the right ring finger (Photo: ESAA Program, Institut Pasteur).**

catch rats or lizards to supplement their diet. The complications of this frequent practice have already been highlighted [4,43–45].

The seasonality of bites differs from that reported in most other studies. The dramatic drop in incidence during the months of May and June seen in our series is not usually observed. This difference cannot be fully explained by the seasonality of ESAA recruitment, which extends from December 2019 to April 2021 inclusive for the three centers in the Far North region, and from May 2020 to April 2021 inclusive for the four centers in the North region (Table 1 and Fig 5). In addition to seasonal recruitment, climatic factors can occasionally alter snake behavior and/or human activities. In the north and far north, the rainy season has been delayed, and the delay has become more pronounced with each passing year. The average annual temperature has increased by 1.9°C compared to the historical average (1979–2015), more in the rainy season than in the dry season [46]. In addition, amount of rain has been reduced by almost half compared to the reference period (1979–2018), but with higher number of rainy days and intensity, leading to significant and frequent flooding. The year 2021 confirms this trend [47]. The months of April and May, the transition period between the dry and rainy seasons, were significantly hotter and drier than in the reference period, delaying the start of the agricultural season.

The frequency of asymptomatic bites in our study (5.4%) was the same as that observed in victims of confirmed *E. ocellatus* bites in eastern Nigeria (4.2%), a region close to where the ESAA study took place [5]. Furthermore, the clinical manifestations described in this study confirm those observed in *E. ocellatus s.l.* species, which include, with similar frequency and severity, pain, edema, bleeding, blistering and, more rarely, tissue necrosis [4,5,7,9,41,43,48–52]. There were no neurological disorders except those due to meningeal or cerebral hemorrhage, which can lead to life-threatening neurological complications, but whose clinical presentation is different from that of elapid SBE.

Cytotoxicity (pain, edema, sometimes necrosis) is the most common manifestation of SBE by vipers, especially *E. ocellatus s.l.* It results from the action of PLA$_2$s and SVMPs, which are abundant in *E. ocellatus* venom [17,18]. The innate immune system identifies molecular motifs from foreign pathogens or abnormal situations [53]. These molecules are recognized by receptors involved in the immune response via the inflammatory response. The molecules present in venom are known as venom-associated molecular patterns (VAMPs), modelled on the pathogen-associated molecular patterns found in pathogens. In addition to the immediate dramatic inflammatory response, VAMPs activate signaling pathways that promote the release of pro-inflammatory mediators, particularly cytokines, which perpetuate the inflammatory cascade [54,55]. This self-stimulation continues even after the neutralization or elimination of the venom, which may explain the inertia of inflammatory manifestations and the low effectiveness of antivenom serums on the evolution of edema. Although a very sensitive indicator of SBE, the high degree of inertia of the edema makes it an inappropriate marker of the evolution of SBE.

Multivisceral failure, simulating sepsis, may lead to the death of the patient independently of other possible causes of death, particularly hemorrhage.

The causes of bleeding in African *Echis* species SBE are multifactorial. The hemorrhagic symptomatology associated with SBE is consistent with the high concentration of SVMPs in *E. ocellatus* venom. SBE by *Echis* spp., including *E. romani*, results in damage at different levels of the coagulation cascade: inhibition of platelet aggregation, thrombocytopenia, prothrombin activation, fibrinogen depletion and fibrinolysis [5,23,24,56].

First, SVMP-induced damage to the vascular endothelium promotes extravasation of blood from blood vessels, contributing to the formation of local edema and distant hematomas resulting from capillary leak syndrome [17,20].

Secondly, the consumption of coagulation factors, in particular due to the presence of prothrombin activators, leads to the depletion of fibrinogen, causing the blood to become incoagulable after a more or less short hypercoagulation time [29,30]. The hypercoagulable phase of the blood has no clinical relevance in *Echis* SBE. Prothrombin activators predominantly produce meizothrombin rather than thrombin, forming an unstable and ephemeral clot that is poorly thrombogenic [23,24].

Thirdly, clot formation is delayed or even reduced by inhibition of platelet aggregation through the action of PLA$_2$s, integrins and C-type lectins [28,29].

Finally, the fibrinolytic action of SVMPs could explain, at least in part, the fibrinolysis observed in some severe SBE [56–60].

Viper venom is eliminated through the kidneys and the gastrointestinal tract [61,62]. After subcutaneous or intramuscular inoculation, the most likely routes for a bite from a snake the size of *E. ocellatus*, the venom becomes undetectable in the bloodstream within 4–7 days. This period is reduced to a few hours after antivenom administration [62,63]. However, the elimination half-life varies according to the molecular weight of the protein. The diffusion and elimination of SVMPs are slower than that of other toxic venom components, suggesting a temporary fixation at the bite site [64].

Warrell et al studied hemostasis in a few late arrival patients whose blood was incoagulable for up to 10 days after the bite [5]. Reid reported incoagulable blood for up to 20 days after being bitten by *E. ocellatus* [65]. A small series of patients who could not receive antivenom, which was not available at the time, was compared with patients treated with the Institut Pasteur polyvalent antivenom *Bitis-Echis-Naja*, then FAV-Afrique (Sanofi Pasteur) [66,67]. Blood parameters (fibrinogen, prothrombin time and thromboplastin time) normalized in 1–3 days in patients treated with antivenom versus 10–14 days in patients without antivenom. Our patients did not benefit from coagulation factor monitoring, only clinical monitoring based on cessation of bleeding. This suggested a rapid normalization of hematologic parameters in most of our patients.

Despite the redundancy of its effects, the direct action of the venom does not explain the prolongation of the hemostasis disorders, since the venom disappears from the organism in less than a week [61,62,64].

Resumption of bleeding after a period of normalization (2–48 hours after antivenom administration) or the prolonged blood incoagulability (up to 20 days after the bite) may be demonstrated by bleeding or detected by an abnormal 20WBCT. These manifestations have been reported in 2–30% of patients with coagulopathy presenting to the hospital [5,9,43,49,51,52,56]. The causes of these hemorrhagic recurrences have not been identified, nor have the causes of the

prolongation of hemostasis disorders in certain patients. Recurrent bleeding and the prolonged blood incoagulability may be related and may share common causes. Several hypotheses –which are not exclusive– have been put forward: failure or delay in the regeneration of coagulation factors, secondary activation of a coagulation factor consumption or fibrinolysis mechanism, maintenance and recirculation of venom in the body [56]. The absence of clotting factor regeneration, which would explain hypofibrinogenemia in particular, is not supported by the presence of liver failure [5,65,67,68]. On the other hand, it may be related to hyperfibrinolysis, which consumes fibrinogen as it is generated [56]. Independent of the fibrinolysis induced by the SVMPs of *E. ocellatus* venom [17], a late secondary hyperfibrinolysis could explain the recurrence of hemostasis disorders [5,57,56,60]. Finally, venom recirculation is an understudied possibility. Venom remains detectable at the injection site for several days after the bite [69].

In any event, death may result from hypovolemic shock, visceral complications such as cerebral hemorrhage, heart failure, acute renal failure, or severe anemia due to blood loss.

It is possible that the initial administration of more vials of antivenom would produce different results in the 15% of patients who experience delayed or recurrent bleeding. However, given the limited availability of antivenom, the protocol used in many countries in sub-Saharan Africa is a sequential approach, in which antivenom is administered in successive doses, with repetition based on clinical progress [70]. This approach therefore requires close monitoring of the patient with a clinical examination every 2 hours to re-administer the antivenom if the clinical presentation does not improve or worsens. However, it is uncertain whether administering a higher dose of antivenom is more effective, as the recirculation of the venom may be due to its sequestration in a part of the body inaccessible to the antivenom [56,69].

Local or extensive necrosis also has a multifactorial cause. It may result from local tissue digestion under the action of venom proteases and $PLA_2$s [71]. SVMPs cause alteration of the endothelial basement membrane, resulting in local ischemia [19]. This also impairs muscle regeneration and leads to tissue fibrosis. In addition, SVMPs degrade the extracellular matrix and are activators of the inflammatory response, which, together with blood extravasation, produces edema reducing local blood perfusion [19]. The formation of extracellular neutrophil traps (NETs) induced by *E. carinatus* venom has been shown to block blood vessels and trap venom toxins at the injection site, promoting tissue destruction and limiting antivenom access to the venom immobilized in the NETs [72]. It has often been reported that edema can cause vascular compression and cessation of distal oxygenation of the bitten limb leading to compartmental syndrome. However, both experimentally and clinically, compartmental syndrome has been shown to play a minor role in necrosis and is rare [73,74]. Finally, we cannot exclude the role of iatrogenic factors such as the application of a tourniquet left in place for too long, or superinfection due to the use of traditional medicines.

## Conclusion

A study of 92 patients bitten by *E. romani*, a species of the *E. ocellatus s.l.* group from SSA, confirmed the frequency and severity of hemorrhagic complications observed in SBE by this species. On the other hand, permanent sequelae due to extensive necrosis were rarer and more moderate in severity than expected.

The population at risk consists of young rural workers who are bitten during agro-pastoral work or on their way to work. Young children are more likely to be bitten on the hand.

In most cases, the administration of antivenom quickly stops the bleeding. However, up to 15% of patients experience prolonged or recurrent bleeding requiring repeated antivenom administration with irregular results. The causes of these prolonged hemorrhagic episodes remain unclear and their treatment is not yet established.

## Supporting information

**S1 Appendix. Management Algorithm Recommended by Cameroonian Ministry of Envenomation Patients (from 33; 34).** The images/clip art appearing in the illustrated panels were hand-drawn by one of the authors (JPC) and are compatible with a CC BY 4.0 license.
(DOCX)

**S2 Appendix. Management Algorithm Recommended by the Cameroon Ministry of Health for Envenomated Patients (from 33; 34).**
(DOCX)

## Acknowledgments

We are indebted to Mr David Benhammou, Ms Anais Perilhou, Mr Fai Karl, Ms Marie Sanchez, Ms Lucrèce Matchim, Ms Lucrèce Eteki, and Mr Mark Ndifon for their contributions to the preparation and follow-up of the ESAA study, which provided us with most of the data used here. We thank the field investigators who made this work possible: Dr Guillaume Gayma et Dr Fadimatou (Mora), Dr Christophe Youmbi et M. Kouli Guidang (Kolofata), Dr Baldagai et Dr Olivier Bito (HR Ngaoundéré), Dr Hans Notaya (HN Ngaoundéré), Dr Hamdja Moustafa (Poli), Dr Ousmana et Mme Ninkouague Jogo (Guider). We are grateful to all the patients who accepted to participate in this study, as well as the regional chief medical officers and the medical staff who participated in this study. We are grateful to Inosan Biopharma for providing the antivenoms free of charge.

We are indebted to Prof. Leslie Boyer and Dr Ellen Einterz for their valuable comments on cases, in particular the serious clinical events, and participation in the scientific committee. We also thank Dr. Luc de Haro, Centre Antipoison of Marseille, France, for his relevant expertise.

## Author contributions

**Conceptualization:** Jean-Philippe Chippaux, Yap Boum, Armand S. Nkwescheu, Fabien Taieb.

**Data curation:** Pierre Amta, Rodrigue Ntone, Gaëlle Noël, Pedro Clauteaux.

**Formal analysis:** Yoann Madec.

**Funding acquisition:** Fabien Taieb.

**Investigation:** Pierre Amta.

**Methodology:** Jean-Philippe Chippaux, Fabien Taieb.

**Project administration:** Rodrigue Ntone, Gaëlle Noël, Pedro Clauteaux, Yap Boum, Armand S. Nkwescheu.

**Resources:** Rodrigue Ntone.

**Supervision:** Jean-Philippe Chippaux, Pierre Amta, Rodrigue Ntone, Gaëlle Noël, Pedro Clauteaux, Yap Boum, Armand S. Nkwescheu, Fabien Taieb.

**Validation:** Jean-Philippe Chippaux, Pierre Amta, Rodrigue Ntone, Yap Boum, Armand S. Nkwescheu, Fabien Taieb.

**Visualization:** Jean-Philippe Chippaux, Yoann Madec, Gaëlle Noël, Fabien Taieb.

**Writing – original draft:** Jean-Philippe Chippaux.

**Writing – review & editing:** Jean-Philippe Chippaux, Yoann Madec, Gaëlle Noël, Yap Boum, Armand S. Nkwescheu, Fabien Taieb.

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
