## [Decision Letter · Decision Letter 0]

Thank you for submitting your manuscript to PLOS Neglected Tropical Diseases. After careful consideration, we feel that it has merit but does not fully meet PLOS Neglected Tropical Diseases's publication criteria as it currently stands. Therefore, we invite you to submit a revised version of the manuscript that addresses the points raised during the review process.

Response to Reviewers
Revised Manuscript with Track Changes

* An unmarked version of your revised paper without tracked changes. You should upload this as a separate file labeled 'Manuscript '.

Shaden Kamhawi

co-Editor-in-Chief

Paul Brindley

co-Editor-in-Chief

**Additional Editor Comments :**

I'm sorry for the slight delay in evaluating your manuscript.

The reviewers' opinion, like mine, suggests that your manuscript meets the quality criteria required for publication by PLOS NTD. However, before doing so, we feel it necessary to answer a few questions, which are mentioned in the evaluation report.

We look forward to receiving your feedback on these points before making a final decision.

**Journal Requirements:**

At this stage, the following Authors/Authors require contributions: Jean-Philippe Chippaux, Pierre Amta, Yoann Madec, Rodrigue Ntone, Gaëlle Noël, Pedro Clauteaux, Yap Boum, Armand S. Nkwescheu, and Fabien Taieb. Please ensure that the full contributions of each author are acknowledged in the "Add/Edit/Remove Authors" section of our submission form.

2) Please ensure that the Title in your manuscript file and the Title provided in your online submission form are the same.

- TM on page: 3.

Potential Copyright Issues:

i) Please confirm (a) that you are the photographer of 3, 8, and 9, or (b) provide written permission from the photographer to publish the photo(s) under our CC BY 4.0 license.

ii) Figure 1. Please (a) provide a direct link to the base layer of the map (i.e., the country or region border shape) and ensure this is also included in the figure legend; and (b) provide a link to the terms of use / license information for the base layer image or shapefile. We cannot publish proprietary or copyrighted maps (e.g. Google Maps, Mapquest) and the terms of use for your map base layer must be compatible with our CC BY 4.0 license.

6) Thank you for stating "The clinical study data have been deposited in a publicly accessible repository" and it is available "at DOI: 10.5281/zenodo.10609046." We noted that the data files are restricted to users with access. Please note that, though access restrictions are acceptable now, your entire minimal dataset will need to be made freely accessible if your manuscript is accepted for publication. This policy applies to all data except where public deposition would breach compliance with the protocol approved by your research ethics board. If you are unable to adhere to our open data policy, please kindly revise your statement to explain your reasoning and we will seek the editor's input on an exemption.

7) Your current Financial Disclosure states receiving a fund. However, your funding information on the submission form indicates no funds. Please ensure that the funders and grant numbers match between the Financial Disclosure field and the Funding Information tab in your submission form. Note that the funders must be provided in the same order in both places as well.

**Reviewers' comments:**

**Key Review Criteria Required for Acceptance?**

**Methods**

-Are the objectives of the study clearly articulated with a clear testable hypothesis stated?

-Is the study design appropriate to address the stated objectives?

-Is the population clearly described and appropriate for the hypothesis being tested?

-Is the sample size sufficient to ensure adequate power to address the hypothesis being tested?

-Were correct statistical analysis used to support conclusions?

-Are there concerns about ethical or regulatory requirements being met?

Reviewer #1: The authors report the results of a clinical study aimed to characterize the symptoms and evolution of patients bitten by Echis romani.

The study was carried out on rural populations in northern Cameroon. Its strengths include formal identification of the snake responsible for envenomation and the standardized longitudinal follow-up of patients in several centres. The study is appropriately designed to assess the objectives, the population is well described and of sufficient size.

No concern about statistical analysis.

Ethical and regulatory requirements met.

Reviewer #2: This is an observational study, based on data collected in a study published elsewhere. Methods for data collection were sufficiently described herein and statistical analysis used simple chi-square tests. Ethical requirements were all met.

The description of the antivenom Inoserp PAN-AFRICA (IPA) did not include data of the venom-neutralizing efficacy tests (potency) of the product. It is known that the in vivo tests provide globally applicable standard metrics of antivenom efficacy, and results from these quality control tests need to be provided by the manufacturer as part of the batch release documentation. This information would be important to discuss the resolution of clinical manifestations of envenoming caused by Echis romani snakes.

Reviewer #3: The study made a careful analysis of the cases of accidents caused by the snake Echis romani, with due ethical follow-up, in rural populations in northern Cameroon.

The antivenom Inoserp PAN-AFRICA proved effective in most treatments, but the persistence of symptoms and mortality could be better explored. Could it be associated with specificity, given the large number of venoms (14 species) used for preparation?

It may be beneficial to consider providing additional information regarding the characterisation of the antivenom, as this could further enrich the discussion.

It is my understanding that there are no problems with the statistics.

**Results**

-Does the analysis presented match the analysis plan?

-Are the results clearly and completely presented?

-Are the figures (Tables, Images) of sufficient quality for clarity?

Reviewer #1: Persented analysis match the plan. Results are clearly and completely presented and figures are of sufficient quality

Reviewer #2: The descriptive analysis showed the demographic characteristics and the clinical manifestations of patients envenomed. In this way, most of results are clearly presented, except for safety data on antivenom treatment, which was omitted. As allergic reactions are known to occur after administration equine derived immunoglobulins, both early and late reactions should have been presented in the study. Moreover, it was mentioned that the patients were evaluated approximately 15 days after hospitalization, to check for possible late reaction.

Reviewer #3: The results and figures are presented in a clear and detailed manner. It has been indicated accidents, with young rural workers, especially children and adolescents, being the most frequently affected. The seasonality of the accidents was assessed, with the highest incidence occurring at the end of the dry season and during the two rainy seasons. The majority of patients exhibited symptoms indicative of both poisoning and haemostasis disorders, manifesting as bleeding during their period of hospitalisation. The antivenom Inoserp™ PAN-AFRICA was administered in most cases, with positive outcomes. However, some patients suffered from prolonged bleeding, and the mortality rate remains significant. A more precise description of the serum might offer valuable insights for the discussion of this condition.

**Conclusions**

-Are the conclusions supported by the data presented?

-Are the limitations of analysis clearly described?

-Do the authors discuss how these data can be helpful to advance our understanding of the topic under study?

-Is public health relevance addressed?

Reviewer #1: This well-documented clinical, biological and therapeutic study underlines the importance of Echis romani envenomation in the area covered by the study and the severity of the prognosis even after well-managed treatment with polyvalent antivenom serum.

Conclusions are well supported by the data. This helpful study opens the way to a better understanding of envenomation and its management and public health relevance.

Reviewer #2: It is noteworthy that seven patients who were not bleeding on arrival at hospital, presented hemorrhage hours after admission and five who had recurrence of bleeding despite the administration of IPA. These 12 patients represent 14% of patients with clinical manifestations of envenomation. In this way, the therapeutic response of patients bitten by E. romani to the scheduled treatment with Inoserp PAN-AFRICA deserved a more detailed analysis.

The authors mentioned that the causes of prolonged bleeding episodes are uncertaint, and one possible cause for these findings include venom recirculation. In this way, would the recommended regimen of initial administration of 2 vials have been sufficient to reverse the hemorrhagic manifestations?

Reviewer #3: It is important to point out the severity of the haemorrhagic complications caused by Echis romani and the limited efficacy of antivenom in cases of prolonged or recurrent bleeding. Although antivenom has shown positive results in initially stopping bleeding, the persistence of haemorrhagic episodes and the high lethality indicate gaps in clinical management and in understanding the mechanisms involved. The conclusion could be more detailed in recommending improvements in treatment management, as well as investigating the causes of prolonged complications.

**Editorial and Data Presentation Modifications?**

Reviewer #1: Accept

Reviewer #2: Figure 6 and 7 are not essential, as data is sufficiently described in the Results section.

Reviewer #3: Minor Revision. The production and characterisation of the serum could be improved through greater detail, in view of the cases of poisoning by Echis romani.

**Summary and General Comments**

Reviewer #1: The authors report the results of a clinical study aimed to characterize the symptoms and evolution of patients bitten by Echis romani.

The study was carried out on rural populations in northern Cameroon. Its strengths include formal identification of the snake responsible for envenomation and the standardized longitudinal follow-up of patients in several centres. The study is appropriately designed to assess the objectives, the population is well described and of sufficient size.

The manuscript is very well written.

This well-documented clinical, biological and therapeutic study underlines the importance of Echis romani envenomation in this area and the severity of the prognosis even after well-managed treatment with polyvalent antivenom serum.

Interestingly, the authors report delayed bleeding. These deserve to be explored by a new study in one or more centres with the possibility of carrying out complementary haemostasis examinations. These would include platelet counts to determine whether thrombocytopenia due to envenomation or associated with the administration of antivenom serum (such as anti-platelet factor 4 antibodies, as in HIT and VITT) could contribute to delayed/persistent bleeding. A dosage of plasma fibrinogen would also be of interest.

Reviewer #2: This study confirms the importance of snakebite envenoming in North Africa, especifically in Cameroon, and highlights the severity of hemorrhagic manifestations that can lead to systemic complications. A specific and efficacious antivenom treatment is essential to reverse this situation, but this issue was poorly analyzed.

Reviewer #3: This study lends further credence to the notion that snakebite envenomation is a significant public health concern in North Africa, particularly in Cameroon. Furthermore, it serves to underscore the gravity of haemorrhagic manifestations, which have the potential to precipitate systemic complications.

The utilisation of antivenom has been demonstrated to be efficacious, and the identified limitations underscore the necessity for additional research to be conducted on the mechanisms of snake envenomation. Furthermore, there is a compelling need for advancements in the specificity and quality of antivenom serum.

PLOS authors have the option to publish the peer review history of their article (what does this mean? ). If published, this will include your full peer review and any attached files.

**Do you want your identity to be public for this peer review?** For information about this choice, including consent withdrawal, please see our Privacy Policy .

Reviewer #1: No

Reviewer #2: No

Reviewer #3: No

**Figure resubmission:****Reproducibility:** To enhance the reproducibility of your results, we recommend that authors of applicable studies deposit laboratory protocols in protocols.io, where a protocol can be assigned its own identifier (DOI) such that it can be cited independently in the future. Additionally, PLOS ONE offers an option to publish peer-reviewed clinical study protocols. Read more information on sharing protocols at https://plos.org/protocols?utm_medium=editorial-email&utm_source=authorletters&utm_campaign=protocols

---

## [Decision Letter · Decision Letter 1]

Dear Dr. Chippaux,

We are pleased to inform you that your manuscript 'Epidemiologic, clinical, and therapeutic aspects of formally identified Echis romani bites in northern Cameroon' has been provisionally accepted for publication in PLOS Neglected Tropical Diseases.

Best regards,

Philippe BILLIALD

Academic Editor

Wuelton Monteiro

Section Editor

Shaden Kamhawi

co-Editor-in-Chief

Paul Brindley

co-Editor-in-Chief

---

## [Editor Report · Acceptance letter]

Dear Dr. Chippaux,

We are delighted to inform you that your manuscript, "Epidemiologic, clinical, and therapeutic aspects of formally identified Echis romani bites in northern Cameroon," has been formally accepted for publication in PLOS Neglected Tropical Diseases.

Best regards,

Shaden Kamhawi

co-Editor-in-Chief

Paul Brindley

co-Editor-in-Chief
